# The hEag1 K^+^ Channel Inhibitor Astemizole Stimulates Ca^2+^ Deposition in SaOS-2 and MG-63 Osteosarcoma Cultures

**DOI:** 10.3390/ijms231810533

**Published:** 2022-09-11

**Authors:** Beáta Mészáros, Agota Csoti, Tibor G. Szanto, Andrea Telek, Katalin Kovács, Agnes Toth, Julianna Volkó, Gyorgy Panyi

**Affiliations:** 1Department of Biophysics and Cell Biology, Faculty of Medicine, University of Debrecen, Life Science Building, Egyetem Ter 1, H-4032 Debrecen, Hungary; 2MTA-DE Cell Biology and Signaling Research Group, Life Science Building, Egyetem Ter 1, H-4032 Debrecen, Hungary; 3Department of Medical Chemistry, Faculty of Medicine, University of Debrecen, Life Science Building, Egyetem Ter 1, H-4032 Debrecen, Hungary

**Keywords:** hEag1 potassium channel, SaOS-2 osteosarcoma cells, Ca^2+^ deposition, Kv10.1

## Abstract

The hEag1 (Kv10.1) K^+^ channel is normally found in the brain, but it is ectopically expressed in tumor cells, including osteosarcoma. Based on the pivotal role of ion channels in osteogenesis, we tested whether pharmacological modulation of hEag1 may affect osteogenic differentiation of osteosarcoma cell lines. Using molecular biology (RT-PCR), electrophysiology (patch-clamp) and pharmacology (astemizole sensitivity, IC_50_ = 0.135 μM) we demonstrated that SaOS-2 osteosarcoma cells also express hEag1 channels. SaOS-2 cells also express to KCa1.1 K^+^ channels as shown by mRNA expression and paxilline sensitivity of the current. The inhibition of hEag1 (2 μM astemizole) or KCa1.1 (1 mM TEA) alone did not induce Ca^2+^ deposition in SaOS-2 cultures, however, these inhibitors, at identical concentrations, increased Ca^2+^ deposition evoked by the classical or pathological (inorganic phosphate, Pi) induction pathway without causing cytotoxicity, as reported by three completer assays (LDH release, MTT assay and SRB protein assay). We observed a similar effect of astemizole on Ca^2+^ deposition in MG-63 osteosarcoma cultures as well. We propose that the increase in the osteogenic stimuli-induced mineral matrix formation of osteosarcoma cell lines by inhibiting hEag1 may be a useful tool to drive terminal differentiation of osteosarcoma.

## 1. Introduction

Osteosarcoma (OS) is the sixth most common cancer in children and adolescents and the 5-year survival rate is low, about 65%, due to its metastatic potential [1]. Therefore, the development of novel treatment strategies is very important for the improvement of the prognosis of OS patients. In the last decade the role of K^+^ channels in cancer growth and progression were intensively studied [2,3,4,5,6] and voltage-gated K^+^ channels were proposed as potential targets in the treatment of multiple tumors [7,8,9] including OS [10,11]. In line with this, the expression of the voltage-gated K^+^ channel Kv7.3 and the Ca^2+^-and voltage-gated KCa1.1 K^+^ channel was reported in both SaOS-2 and MG-63 osteoblast-like osteosarcoma cells [12,13,14]. Moreover, the KCa1.1 and Kv7.3 currents were also identified in these cells using the patch clamp, and the pharmacological sensitivity of the KCa1.1 current to TEA, charybdotoxin and paxilline [13], and that of the Kv7.3 current to linopirdine and XE991, were demonstrated [14].

The voltage-gated hEag1 K^+^ channel (Kv10.1) is normally expressed in the brain and myoblasts, but its ectopic expression was described in a wide range of cancer and tumor cells [6,15,16,17]. Eag1 may participate in the regulation of the cell cycle and cancer/tumor progression [4,18,19,20]. The overexpression of hEag1 was described in solid tumors and in human osteosarcoma and osteosarcoma cell lines as well [21,22,23,24]. Although the expression of hEag1 in SaOS-2 and MG-63 osteosarcoma cell lines was shown using molecular biological techniques (RT-PCR, Western blot, and immunohistochemistry), the Eag1 currents were not identified in these cells using electrophysiology [11,19,21,22,23].

Both SaOS-2 and MG-63 cell lines serve as a useful experimental model to study the mineral matrix production of osteosarcoma cells, e.g., SaOS-2 cells share some osteoblastic features and can fully be differentiated in the same way as the osteoblastic cells naturally do [25,26]. Although activation of the appropriate intracellular signaling pathways is key to regulate osteogenesis and mineral matrix production [27], it seems that the electrophysiological properties of the cells also play a crucial role in this process [28,29,30]. In line with this, we have shown recently that mineral matrix production in chorion-derived mesenchymal stem cells depends on the activity of the hHv1 proton channel [31]. Several lines of evidence suggest that modulation of the K^+^ conductance of the membrane alters Ca^2+^ deposition, mineral matrix formation and expression of osteogenic genes in osteosarcoma cell lines. For example, inhibition of Kv7.3 K^+^ channels by linopirdine or XE991 increased matrix mineralization during osteoblast differentiation [14] and inhibition of the KCa1.1 K^+^ channel by TEA (0.1–3.0 mM) significantly increased mineralization in primary osteoblasts [13]. Moreover, the application of charybdotoxin, a peptide blocker of the KCa1.1 channels, increased vitamin D-induced osteocalcin secretion in MG-63 cells [32]. Inhibition of Eag1 expression by siRNA or shRNA decreases proliferation and migration in SaOS-2 and GM-63 [11,19], however, the effects produced by the blockade of Eag1 on Ca^2+^ deposition have not been investigated yet.

Based on these, we designed experiments to test the functional expression of hEag1 K^+^ channels in SaOS-2 osteosarcoma cells and to determine if pharmacological modulation of the channel may play a role in the regulation of osteogenic differentiation. Using molecular biology (PCR), electrophysiology (patch-clamp) and pharmacology (astemizole block), we showed that hEag1 is expressed in SaOS-2 along with the KCa1.1 channel. Astemizole (in 2 µM) significantly increased/enhanced Ca^2+^ deposition, a marker of mineral matrix production, of SaOS-2 and MG-63 cell cultures when osteogenic differentiation was induced either by the classic or the pathological pathway. This effect was observed if astemizole was applied within the first 24 h of osteogenic induction. Our results suggest that inhibition of Eag1 K^+^ channels may be a useful strategy to regulate osteoblast differentiation.

## 2. Results

### 2.1. hEag1 (Kv10.1) Is Expressed in SaOS-2 and MG-63 Cells

Using a reverse transcriptase PCR(RT-PCR), we have confirmed earlier studies that the transcript of hEag1 can be identified in SaOS-2 cells (Appendix A) [11,19,21,22,23]. The functional expression of hEag1 channels in the plasma membrane of SaOS-2 cells was confirmed using the patch clamp technique (Figure 1). One of the biophysical hallmarks of the hEag1 current is the very prominent slowing of the activation kinetics of the current evoked from negative holding potentials. Figure 1A shows that the current evoked from −120 mV holding potential requires more than 100 ms to reach the peak as compared to the less than ~20 ms when the currents were evoked from −60 mV holding potential. Consistent with this, the time constant for the activation kinetics of the current was smaller when evoked from −60 mV (τ_act_ = 12.76 ± 1.62 ms, *n* = 5) than from −120 mV (τ_act_ = 56.47 ± 6.13 ms, *n* = 5). This phenomenon resembles the Cole-Moore shift described for the K^+^ current in squid axons and observed in Shaker-type K^+^ channels [33,34], although the molecular mechanism of the very prominent slowing of the activation kinetics is different in hEag1 [35]. To confirm the molecular identity of the channel, additional pharmacological experiments were conducted using astemizole as this compound displays a characteristic open-channel block of hEag1 (Figure 1B). The open-channel block is reflected in an apparent inactivation-like decay of the current at high (2 μM) drug concentration. To construct the concentration–response relationship of the astemizole block (Figure 1C), the remaining current fractions (RCF) were determined from the equilibrium block at the end of the 3000-ms-long depolarizing pulses. Figure 1C shows that the IC_50_ of astemizole, determined by fitting the Hill equation to the concentration–response relationship, is ~135 nM, which corresponds well to the published values in the literature [36]. It is important to note that the concentration–response relationship could be fitted assuming an astemizole-insensitive current fraction that corresponds to ~8% of the total current (a = 0.08, see Methods). This may mean the presence of other astemizole insensitive channels in the membrane of SaOS-2. In line with this, the presence of the KCa1.1 K^+^ current was demonstrated as a paxilline-sensitive whole-cell current (Figure 1D). In these experiments, the cells were depolarized to +100 mV and the pipette-filling solution contained 4.5 μM free Ca^2+^ to allow maximal activation of the KCa1.1 current. The fraction of the whole-cell current inhibited by 1 μM paxilline was 0.19 ± 0.04 (*n* = 9). Thus, the dominant component of the whole-cell current is paxilline-insensitive even if the recording conditions were optimized for KCa1.1 (Figure 1D). Based on this, we concluded that under the experimental conditions used for Figure 1A–C the dominant current is hEag1 (i.e., depolarization to +50 mV and 0 nominal free Ca^2+^ concentration), and as noted above, the astemizole-insensitive current fraction was negligible (a = 0.08).

Since the pharmacological profile of hEag1 overlaps with hErg1(Kv11.1) [37,38,39], ruling out the contribution of hErg1 to the SaOS-2 currents was necessary for the validation of the pharmacological experiments (see below). We did not find the expression of hErg1 transcripts using QPCR or hErg1 proteins using Western blotting (Appendix A) in either undifferentiated SaOS-2 or MG-63 cells or in the negative control (peripheral blood mononuclear cells). Positive controls using hErg1-transfected HEK cells or HEK cells expressing hErg1 in a stable manner showed the appropriate mRNA and protein expression (Appendix A). To confirm these molecular biology data, we carried out a set of electrophysiological experiments to address if hErg1 current is expressed in SaOS-2 (Figure 2). Figure 2A shows that using the hErg1-optimized intra- and extra-cellular solutions proper hErg1 current can be recorded in an HEK-293 cell line expressing hErg1 in a stable manner. Similar currents were recorded in four additional cells (see also in [40,41]). Under the same experimental conditions (ion composition and voltage protocol), a robust hEag1 current can also be recorded in an HEK-293 cell transfected with the Kv10.1 gene (Figure 2B). Similar currents were recorded in three additional cells. Thus, our positive controls confirm the ability of recording hErg1 using the voltage protocol and ion composition of the solutions optimized for hErg1 (Figure 2A) and gives information about the shape of the hEag1 current driven by the hErg1 voltage protocol (Figure 2B). The hErg1 protocol-driven hErg1 current lacks the slow component of the decay at −40 mV following the depolarization to +20 mV, as the hEag1 current deactivates instantly upon the repolarization to −40 mV. Panel C in Figure 2 indicates that the whole-cell current evoked by the hErg1 voltage protocol in an undifferentiated SaOS-2 cell is qualitatively similar to the hEag1 current in Panel B (Kv10.1-transfected cell), i.e., it lacks the slow decay of the current at −40 mV following the depolarization to +20 mV. Rather, the whole-cell current drops instantaneously upon repolarization to −40 mV, like the pure Kv10.1 current in transfected HEK-293 (Figure 2B). Similar currents were recorded in *n* = 9 SaOS-2 cells driven by the hErg1-optimized voltage protocol and using an hErg1-optimized ion composition of the solutions. Based on these, we propose that Kv11.1 (hErg1) current is not expressed in an undifferentiated SaOS-2 cell at a detectable level using the patch clamp.

### 2.2. Ca^2+^ Deposition Is Enhanced in the Presence of K^+^ Channel Inhibitors

Osteogenic differentiation of SaOS-2 and MG-63 cells were induced using the “classical” and the “pathological” pathways. For the classical induction pathway, we applied dexamethasone, ascorbic acid, β-glycerophosphate and vitamin D3 treatment, whereas pathological mineralization was induced using inorganic phosphate (P_i_) (see Materials and Methods). In vitro studies traditionally use alizarin red staining as a standard for the detection and quantification of Ca^2+^ deposition in the mineral matrix, which is an indicator of osteoblast activity during differentiation and mineralization [13,42,43]. Accordingly, mineralization was qualitatively followed using alizarin red staining of the cultures and was quantitatively measured by extracting the calcium deposit–alizarin red complexes (see Materials and Methods). Figure 3 shows SaOS-2 cell cultures in the absence and presence of the mineralization-inducing cocktail for the classical pathway. As a positive control for the dependence of Ca^2+^ deposition on K^+^ channel function we applied a general K^+^ channel blocker tetraethyl ammonium (TEA). TEA was reported earlier to enhance mineralization at a 0.1–3 mM concentration range in primary osteoblasts, where TEA blocks the large conductance Ca^2+^-activated K^+^ channel, KCa1.1 [13]. We added TEA (in PBS) to the mineralization-inducing cocktail at 1 mM concentration and analyzed the Ca^2+^ deposition in the cultures (Figure 3). The images in Figure 3 show that bone nodules appear on Day 3 following osteogenic induction (column II) and the density, and the size of bone nodules increases as the culture progresses to Day 4 and Day 5, the red stain that indicates Ca^2+^ deposits becoming dominant. Figure 3 also indicates an increased number of osteogenic nodules with a dense red color in the presence of 1 mM TEA (column III.) regardless of the day of the assay following mineralization induction (Day 3, 4 and 5; Figure 3). We interpret the increase in the Ca^2+^ deposits as an indicator of an enhanced mineral matrix production of SaOS-2 cells when K^+^ channels are inhibited by 1 mM TEA.

To analyze the contribution of hEag1 to the mineral matrix production of osteosarcoma cells, we used the hEag1 blocker astemizole. Based on the electrophysiological results (Figure 1) we used astemizole in 2 µM concentration (~10× IC_50_) in all experiments. As a positive control, we used 1 mM TEA treatment in all parallel experiments (Figure 4 and Figure 5). When we compared Ca^2+^ deposition in the presence of the different inhibitors, the TEA-treated differentiated cells were compared with the PBS-treated differentiated cells (vehicle control), and the astemizole-treated differentiated cells were compared with the 0.1% *v*/*v* DMSO-treated differentiated cells (vehicle control). The analysis of the Ca^2+^ deposition was carried out on Day 3 and Day 4 after the induction of mineralization (Figure 4 and Figure 5). Normalized Ca^2+^ deposition was calculated from the absorbances of the different samples upon extraction of the calcium deposit–alizarin red complexes (see details in Figure 4 and Figure 5 and in the Materials and Methods).

Astemizole treatment enhanced Ca^2+^ deposition of SaOS-2 cells induced by the classical osteogenic pathway (Figure 4). Normalized Ca^2+^ deposition was significantly higher both in the TEA-treated samples, and in the astemizole-treated differentiated SaOS-2 cells as compared to their respective vehicle controls (*p* < 0.05, one-way RM ANOVA) regardless of the day of the examination (Day 3: Figure 4A and Day 4: Figure 4B). There was no significant difference in the normalized Ca^2+^ deposition measured in the presence of 1 mM TEA and 2 µM astemizole (*p* > 0.05). We also tested the effect of astemizole on the inorganic phosphate (Pi)-induced mineralization of SaOS-2 cells (Figure 5 and Appendix A). The same set of experiments was carried out as above, and data obtained in the presence of the blockers were compared with vehicle controls. Astemizole in the 2 µM concentration significantly increased (*p* < 0.05, one-way RM ANOVA) Ca^2+^ deposition of SaOS-2 induced by the pathological induction pathway (Figure 5), both on Day 3 and Day 4. This effect was similar to that of 1 mM TEA (*p* > 0.05). Neither the 1 mM TEA nor the 2 µM astemizole affected Ca^2+^ deposition on either Day 3 or Day 4 in the of absence of osteogenic induction (Appendix A). In all experiments above (Figure 4 and Figure 5), ion channel blockers were present in the inducing medium from the beginning of the osteogenic induction. Interestingly, neither astemizole nor TEA had any effect on Ca^2+^ deposition when we applied these drugs 24 h after the osteogenic induction (Figure 6). The readouts of normalized Ca^2+^ deposition in these experiments were on Day 4 and Day 5 to provide identical exposure of the cultures to TEA and astemizole as in Figure 4 and Figure 5.

We extended our study to the MG-63 osteosarcoma cells as well to address if the effect of astemizole was cell type-specific. Although both SaOS-2 and MG-63 can produce mineral matrix, the osteoblastic features of these cells are different [26]. Ca^2+^ deposition was less effective, and its production required extended culturing time in MG-63 (5 days for SaOS-2 vs. more that 20 days for MG-63, Appendix A). Regardless of these differences, the effects of astemizole are qualitatively similar to those obtained for SaOS-2 cells, i.e., 2 µM astemizole potentiated Ca^2+^ deposition in MG-63 cultures as well (Appendix A). The readouts of the Ca^2+^ depositions were obtained at several, extended time points since the kinetics of the mineral matrix production of MG-63 cells was significantly slower than mineralization of SaOS-2 cells using either physiological-like induction (Appendix A) or a pathological one (Appendix A).

### 2.3. TEA and Astemizole Do Not Induce Cytotoxicity

Necrotic and apoptotic cells may serve as nuclei for calcification [44]. To rule out that TEA and astemizole increase Ca^2+^ deposition though a non-specific manner by inducing apoptosis/necrosis, we tested the effect of these K^+^ channel blockers on cellular cytotoxicity/cell viability using a combination of three methods: LDH release, MTT and SRB assays (Figure 7). The MTT reduction assay is based on the conversion of MTT (thiazolyl blue tetrazolium bromide) to formazan by NAD(P)H-dependent oxidoreductase enzymes of viable cells only. The amount of formazan directly correlates to the number of metabolically active cells [45]. Lactate dehydrogenase release (LDH release) assay is based on measuring the activity of cytoplasmic LDH enzyme released by damaged cells. The enzyme is rapidly released into the cell culture supernatant upon apoptosis, necrosis, and the plasma membrane damage. LDH activity can be quantified, and the amount of LDH is directly proportional to the number of dead or damaged cells [46]. The sulforhodamine B (SRB) assay is based on the ability of sulforhodamine B to bind electrostatically, in a pH-dependent manner, to basic amino acid residues of proteins in trichloroacetic acid-fixed cells. Thus, the assay reports on protein abundancy in the samples, which is expected to decrease when cell growth is compromised. The SRB assay was used successfully to measure drug-induced cytotoxicity over a wide range of cell concentrations and cytotoxic effects [47,48].

All these assays are colorimetric, results are presented in Figure 7 as normalized absorbances (in %), the K^+^ channel blockers were added to non-differentiated cell cultures. Figure 7A shows that osteogenic induction by the classical pathway increased significantly the LDH release as differentiation is accompanied by increase in cell death [44]. This treatment served as a good positive control for the interpretation of the assay. The amount of LDH enzyme released in the supernatant of astemizole-treated or TEA-treated cells was statistically the same as compared to 0.1% *v*/*v* DMSO and PBS controls, respectively (Figure 7A), regardless of the day of the examination (Day 3 and Day 4). The MTT assay on Day 3 and Day 4 show a similar pattern, however, in this case a decrease of the normalized absorbance (%) reports on cytotoxicity (Figure 7B). Figure 7B shows that osteogenic induction is associated with a significant decrease in the normalized absorbance in the MTT assay which is consistent with a decrease in the viable cells as differentiation progresses. As with the LDH-release assay, the MTT assay reports similar normalized absorbance of astemizole-treated or TEA-treated cells as compared to DMSO and PBS controls, respectively (Figure 7B), regardless of the day of the examination (Day 3 and Day 4). There was, however, a significant difference between the PBS (ctrl+PBS) and the DMSO controls (ctrl+DMSO) on Day 3. We do not know the source of this difference. Figure 7C shows that the results of the SRB protein assay mirrored those of the MTT assay, osteogenic induction decreased the normalized absorbance on both Day 3 and Day 4. Neither TEA nor astemizole decreased the protein content of the samples on either Day 3 or Day4 as reported by the SRB assay. In summary, our results suggest that neither 2 µM astemizole nor 1 mM TEA had cytotoxic effect on SaOS-2 cells.

## 3. Discussion

Using molecular biology tools, several previous publications indicated the expression of hEag1 in SaOS-2 and MG-63 osteosarcoma cell lines [11,19,21,22,23]. We have confirmed these findings and demonstrated the presence of hEag1 transcripts in SaOS-2 cells using molecular biology (RT-PCR and QPCR). However, to our knowledge, we are the first to demonstrate the functional expression of the channel using a whole-cell patch clamp. We argue that most of the whole-cell currents in SaOS-2 cells is hEag1, based on the demonstration of two hallmarks of this current. First, the hEag1 current is characterized by a very prominent slowing of the activation kinetics from a hyperpolarized holding potential [34]. This phenomenon is demonstrated clearly for the whole-cell currents recorded in SaOS-2 cells in Figure 1. Another hallmark of hEag1 is the characteristic open channel block of the current by astemizole. To demonstrate the inhibition of the hEag1 by astemizole we used a −60 mV holding potential where activation kinetics of the current is fast (Figure 1) and consequently block equilibrium develops fast, reaching steady-state within 3000 ms at various concentrations of the drug (Figure 1B). Accordingly, the currents at the end of the 3000-ms depolarization measured in the presence and absence of various concentrations of astemizole [36] were used to construct the concentration–response curve. The IC_50_ of 0.135 µM for inhibition of the whole-cell current by astemizole in SaOS-2 cells is in good agreement with the previously published data (~200 nM; [4]). Based on the biophysics and pharmacology we conclude that hEag1 channels are functionally expressed in SaOS-2 cells.

The block of the whole-cell currents was not complete even at very high astemizole concentration (e.g., 10 µM, Figure 1C), which may indicate the presence of other conductances in the membrane of SaOS-2 cells (in addition to a small residual Eag1 current at high astemizole concentration, see [36,49]). It has been shown earlier, using electrophysiology, that KCa1.1 channels are expressed in SaOS-2 cells [12]. Our experiments confirmed this, that 1 µM paxilline inhibited the whole-cell currents when conditions appropriate for recording the KCa1.1 current were applied. The magnitude of the paxilline-sensitive current in SaOS-2 under physiological conditions, however, is overestimated based on the data in Figure 1D, since this current was recorded at +100 mV test potential and at 4.5 μM free Ca^2+^ concentration (Figure 1D). The transcript of Kv7.3 channels and the expression at protein level (Western blot) were also reported in SaOS-2 along with the recording of Kv7.3 currents in MG-63 [14]. Kv7.3 channels produce a non-inactivating K^+^ current, which is indistinguishable from the currents shown in Figure 1. In the absence of high affinity selective blockers of Kv7.3 [50] we did not investigate the presence of this conductance in detail. Regardless, the contribution of the astemizole-insensitive current to the total whole-cell current is ~8% (Figure 1C) which indicates the dominance of the hEag1 current in SaOS-2 cells.

hEag1 (Kv10.1) and hErg1 (Kv11.1) channels belong to two closely related families of Kv channels, and both are astemizole-sensitive [36,51]. The expression of the hErg1 channel was described at transcript and protein levels in the MG-63 osteosarcoma cell line [52]. Our results obtained using QPCR and Western blot techniques (Appendix A.) did not confirm the expression of hErg1 either in MG-63 or SaOS-2 cells, and our conclusion is supported by appropriate positive and negative controls. Moreover, our electrophysiological measurements did not show the expression of an hErg1-like current in SaOS-2 when the hErg1 specific voltage protocol and pipette-filling/bath solutions were used (Figure 2). Based on these, we conclude that most of the K^+^ current in SaOS-2 is hEag1 and that the pharmacological effects of astemizole (see below) can be attributed to the inhibition of hEag1.

It was reported earlier that dead (apoptotic or necrotic) cells can form nuclei for calcification [44,53]. Thus, the potential of astemizole to induce cellular toxicity is critical for the interpretation of Ca^2+^ deposition. Osteogenic differentiation includes cell death, so the examination of cytotoxicity is possible in the undifferentiated cells only. Figure 7 indicates this clearly, that differentiation induction resulted in an increase in LDH release and decrease in the formazan production (MTT assay) and in the protein content of the cells as well (SRB assays). This allows differentiation induction to be used as a good positive control for the validation of cell viability or cytotoxicity assays [54,55,56]. Our results clearly demonstrate that neither the hEag1 inhibitor astemizole (2 µM) nor TEA (1 mM) increased LDH release, or reduced the readout of the MTT assay, moreover, SRB assay was also insensitive to both astemizole and TEA treatment. In these latter experiments we applied the K^+^ channel blockers to SaOS-2 cultures in the absence of differentiation induction. hEag1 inhibition is known to reduce the proliferation of cancer cells [6,57,58,59,60,61,62] and thus, astemizole was expected to reduce the cell proliferation. We explain the lack of astemizole’s effect on the MTT assay by the high-density culture used in the current study where contact inhibition is already significant [63] and further reduction of cell proliferation cannot be induced by the block of hErg1. Based on these, our conclusion is that K^+^ channel inhibitors TEA and astemizole do not influence cytotoxicity and/or necrotic nuclei formation and thus, any effect on Ca^2+^-deposition may rather be related to the block of the conductances rather than non-specific effects on cell viability.

Several previous reports indicate that the expression of several osteogenic markers (Runx-2, Osterix transcription factors, osteocalcin expression, alkaline phosphatase activity) run parallel or is followed by calcium deposition in various cell cultures when osteogenic differentiation is induced, and that the measurement of Ca^2+^ deposition in the cultures using the alizarin red staining is a widely accepted approach to illustrate the extent of osteogenic differentiation and calcified bone matrix formation [18,64,65,66,67]. Accordingly, we used the alizarin red staining method to measure the effect of ion channel inhibitors on the endpoint of osteogenic differentiation, i.e., calcified bone matrix formation. We showed that 1 mM TEA and 2 µM astemizole significantly increased the Ca^2+^ deposition in SaOS-2 cell cultures (Figure 3, Figure 4 and Figure 5). TEA at 1 mM concentration blocks ~90% of KCa1.1 channels (IC_50_ ~0.1–0.2 mM, [68,69,70]), whereas it inhibits a negligible fraction of the hEag1 current only (10 mM TEA blocked about 10% of the hEag1 current at +50 mV [71]). Therefore, we attribute the increase in Ca^2+^ deposits in SaOS-2 cultures in the presence of 1 mM TEA to KCa1.1 inhibition, as it was shown in MG-63 osteoblast-like osteosarcoma cells [12,13,32], primary osteoblasts [13] and bone-marrow-derived human mesenchymal stem cells during osteogenic differentiation [14]. The TEA affinity of Kv7.3 is above 30 mM [72] and there is no indication in the literature about Kv7.3 sensitivity to astemizole, therefore, we can exclude the contribution of the Kv7.3 inhibition to the results obtained where Ca^2+^ deposition was studied. The block of Kv7.3 by linopirdine or XE991 was shown to increase the matrix mineralization in MG-63 cells [14], so the literature agrees that K^+^ channel inhibition leads to an increase in the mineral matrix formation.

The concentration of astemizole used in our study (2 µM) was set to produce comparable inhibition of the hEag1 channels (~90%) to that of KCa1.1 channels by 1 mM TEA. The increase in Ca^2+^ deposition was similar in the presence of 1 mM TEA and 2 µM astemizole (Figure 4 and Figure 5), regardless of the method of osteogenic induction. This may indicate that K^+^ currents conducted by Eag1 and KCa1.1 channels regulate Ca^2+^ deposition in a similar manner is SaOS-2 cultures albeit the Eag1 current is larger than KCa1.1 current as reported by the electrophysiological assay. Regardless, the blocker sensitivity of the Ca^2+^ deposition is consistent with the hypothesis that the inhibition of the K^+^ conductance is responsible for the observed effects. Moreover, we showed that K^+^ channel inhibition increased Ca^2+^ deposition only if the blockers were present in the first 24 h following induction. The application of either TEA or astemizole past that time window did not influence the calcified matrix production regardless of the method of induction (Figure 6). This may indicate that the K^+^ conductance of the membrane modifies early signaling events upon osteogenic induction (see below). We also demonstrated the sensitivity of Ca^2+^-deposit production to astemizole in the hEag1-expressing cell line MG-63 [19,21,22,23]. The effect of astemizole treatment was observed at a later time point in MG-63 cells than in SaOS-2 as the overall kinetics of mineralization is significantly slower in MG-63 [26]. Thus, the effect of astemizole is not cell-line specific, but consistent with the expression of hEag1.

Zhang et al. reported that the silencing of hEag1 channels decreased osteogenic differentiation with reduction of mineral precipitation and osteocalcin expression in human bone-marrow-derived MSC [73]. The distinct effect of the hEag1 inhibition in SaOS-2 and gene silencing MSCs may originate in the cell type studied, or perhaps, in that gene silencing inhibits the expression of hEag1 protein, and thus the non-canonical functions of the channel as well [4,39,74], whereas astemizole inhibits the canonical function of the channels only. Several publications describe the role of hEag1 channels in molecular processes where the bifunctional feature of these channels is an important factor [4,39,74,75,76]. Interestingly, mRNA expression of hEag1 increases significantly during mineralization in SaOS-2 cells (Appendix A), which may support the role of this channel beyond the 24 h window defined by astemizole (~K^+^ channel block) sensitivity of Ca^2+^ deposition. Moreover, hEag1 mRNA expression was higher in mineralization-induced cultures when the culturing medium also contained 2 µM astemizole (Appendix A). We could not confirm hEag1 functional activity in the plasma membrane of differentiated SaOS-2 cells using the patch clamp due to the physical properties of the mineralized tissue.

The critical link between K^+^ channel inhibition and enhanced Ca^2+^ deposition in SaOS-2 and MG-63 cultures may be the regulation of the membrane potential of the cells. For example, K^+^ channel inhibition-mediated depolarization may alter the activity of the Na^+^ and P_i_ transport across the cell membrane through the Na^+^-dependent P_i_ cotransporter PiT1 [77]. Increased intracellular P_i_ level alters the transcription of key regulators of the mineral matrix formation, such as STC1, Runx2, osteocalcin, alkaline phosphatase, integrin binding sialoprotein, osteopontin, osterix, etc. [77,78,79,80,81]. Consistent with this, we found that the inhibition of either KCa1.1 or hEag1 increases Ca^2+^ deposition induced by the differentiation cocktail mimicking physiological mineralization. At the same time P_i_-induced Ca^2+^ deposition (pathological mineralization) was also increased by the K^+^ channel inhibitors. Based on this, P_i_-induced pathological mineralization in vascular smooth muscle cells [81,82,83] may also be influenced by the K^+^ channel inhibition. However, it is important to clarify that in the absence of osteogenic induction, inhibition of the ion channels does not influence Ca^2+^ deposition by SaOS-2 cells (Appendix A), i.e., K^+^ channel inhibition modulates, but does not induce itself to cause osteogenic differentiation and calcified matrix production.

hEag1 participates in the regulation of the proliferation of osteosarcoma cells [19], and this K^+^ channel has already been listed as a potential target in the cancer therapy [21,22,58,59,60,61,76,84,85,86,87,88,89]. Our results highlight that the inhibition of Eag1 also modulates Ca^2+^ deposition into the mineral matrix of SaOS-2 and MG-63 cells. Thus, Eag1 inhibition, using a yet-to-be-developed highly selective and potent inhibitor, may have double benefits in the treatment of osteosarcoma; on one hand the inhibition of the osteosarcoma cell proliferation and on the other hand the stimulation of Ca^2+^ deposition into a terminal state of the bone matrix, which may be a new potential therapeutic strategy in cancer therapy.

## 4. Materials and Methods

### 4.1. Cell Culture and Differentiation and Ion Channel Expression

SaOS-2 and MG-63 osteosarcoma cells were cultured in DMEM and supplemented with 10% FBS, 100 U/mL penicillin, 100 µg/mL streptomycin and 15 L-glutamine at 37 °C in a humidified atmosphere of 5% CO_2_ (growth medium). For the induction of mineralization, there were two different methods: the “classic inducing medium” was supplemented with dexamethasone (0.1 µM), ascorbic acid 2-phosphate (50 µg/mL), β-glycerophosphate (10 mM) and vitamin D3 (50 nM) [18,31]. Pathological mineralization was induced by inorganic phosphate (1.5–3 mmol/L; pathological-inducing medium) [31,82]. Cultures were initially seeded at a cell density of 580 cells/mm^2^ (high density culture) in all experiments. Cells were seeded in a growth medium and allowed to attach for 24 h, then differentiation was initiated by adding an osteogenic medium (Day 0).

HEK 293 cells and HEK cells with stable expression of the hErg (Kv11.1) channel were cultured under standard conditions as described elsewhere [40,41]. HEK 293 cells were transiently transfected with the gene encoding the Kv10.1 channel (KCNH1 gene, OriGene Technologies), using Lipofectamine 2000 (Invitrogen, Carlsbad, CA, USA) as described elsewhere [90].

### 4.2. RNA Extraction and cDNA Synthesis

The total RNA was extracted from SaOS-2 and MG-63 cells using GenElute Mammalian Total RNA Miniprep Kit (Sigma, St. Louis, MI, USA) according to the manufacturer’s instructions. The RevertAid H Minus First Strand cDNA Synthesis Kit (Thermo Scientific^TM^ K1632, Thermo Fisher Scientific, Waltham, MA, USA) was used for the reverse transcription according to the manufacturer’s instruction. NAC (no amplification control) for each RNA sample was produced by setting up the RT reverse transcription reaction as usual but omitting the reverse transcriptase. The effectiveness of RNA isolation and cDNA synthesis were controlled with amplifying GAPDH mRNA using primers from the RevertAid H Minus First Strand cDNA Synthesis Kit (Thermo Scientific ^TM^ K1632, Thermo Fisher Scientific, Waltham, MA, USA) according to the manufacturer’s instructions.

### 4.3. Reverse Transcription PCR (RT-PCR) and Quantitative PCR (QPCR)

PCR amplification of cDNA was performed in a final volume of 20 µL containing 3 µL diluted (10×) cDNA solutions, 5 µL 5X Phusion HF High-Fidelity Puffer (ThermoScientific), 0.5 µL 10 mM dNTP, 0.5 µL Phusion High-Fidelity DNA Polymerase (ThermoScientific), and 10–10 pmol of each primer and nuclease-free water. The sequences and the positions of the primers were for KCNH1 (NM_002238.3): 5′-TCT GTC TAC ATC TCC TCG TT-3′ (1431–1450) and 5′-CCA TTA CTC GCT CAC TCA-3′ (1701–1684) and for KCNMA1 (NM_001014797.2 NM_001014792.2): 5′-AGC AGC CGT CAA CAC TAT-3′ (2291–2308) and 5′-AGT TAT GAA GCG TCT CCC-3′ (2701–2684). The amplification conditions were the same for both ion channel genes. PCR cycling conditions were the following: 98 °C for 30 s followed by 35 cycles of 98 °C for 20 s, 56 °C for 30 s and 72 °C for 30 s, with a final extension step of 72 °C for 10 min. All PCR productions were visualized on 1.5% agarose gels by staining with ethidium bromide. Quantification of hEag1 mRNA expression was examined using the StepOnePlus™ Real-Time PCR System, TaqMan Gene Expression Master Mix (Applied Biosystems, 4369016), UPL Probe 66 (Roche, Basel, Switzerland, 4688651001). GAPDH gene (Thermo Fisher Scientific, 4331182; hs02786624_g1) was used as the internal standard. The quantification of hErg1 transcripts was measured using the same instrument, but the QPCR reaction contained LightCycler 480 SYBR Green I Master Mix (Roche, 04887352001), ROX Reference Dye (Invitrogene, Waltham, MA, USA, 12223012). The primer sequences were as follows: hErg1 forward 5′-CAA AGT GGA AAT CGC CTT CT-3′, hErg1 reverse 5′-ACC ACA TCC ACC AGA CAT AGG-3′. The internal standard was actin (ACTB; primer sequences: forward 5′-CAT GGC TGG GGT GTT GAA-3′; reverse 5′-GAG GAG CAC CCC GTG CT-3′).

### 4.4. Western Blot

Cells were dissolved in a 400 µL protein lysis buffer (50 mM Tris, 150 mM NaCl, 5 mM EDTA, 1% Triton X-100), which was completed with 60 µL Protease Inhibitor Cocktail (Roche, Complete Mini, 1 tablet/10 mL) and 4 µL 100 mM PMSF (phenyl-methane-sulfonyl fluoride) at 4 °C. Lysates containing 50–100 µg protein were mixed with SDS puffer and incubated at 98 °C for 10 min. Protein was electrophoresed on 7% and 10% SDS–polyacrylamide gels and electroblotted onto a PVDF membrane. The blot was first blocked with 5% BSA in 0.2% Twin-PBS (TPBS) for 2 h. Then, anti-hErg1 (hERG1 CT, pan–polyclonal antibody, Di.V.A.L. Toscana SRL (#DT-552), Rabbit pAb) and anti-actin (Sigma, A2066, produced in rabbit) antibodies against the appropriate proteins, diluted in 5% BSA TPBS in 1:1000 and 1:8000 were added, and incubated at 4 °C overnight. After the washing, the membranes were incubated with a highly cross-adsorbed peroxidase antibody (diluted in 1:3000; Sigma, SAB3700853) at room temperature for 1 h. Each step was followed by 3 × 15-min washes in TPBS. Immunoreactivity was determined by a chemiluminescent reaction (SuperSignal™ West Pico PLUS Chemiluminescent Substrate, Thermo Scientific™, 34580).

### 4.5. Electrophysiology and Chemicals

Electrophysiology measurements were carried out using the patch clamp technique in voltage-clamp mode. Whole-cell currents were recorded in SaOS-2 cells. For the recording of ionic currents, an Axopatch 200B amplifier and a Digidata 1440 digitizer was used (Molecular Devices, Sunnyvale, CA, USA). Micropipettes were pulled from GC 150 F-15 borosilicate capillaries (Harvard Apparatus, Holliston, MA, USA) resulting in 3- to 5-MΩ resistance in the bath solution. The bath solution generally contained 145 mM NaCl, 5mM KCl, 10 mM HEPES, 5.5 mM glucose, 2.5 mM CaCl_2_, 1 mM MgCl_2_, pH 7.35. The measured osmolarity of the extracellular solution was about 300 mOsm/L. To measure hEag1 currents, the pipette solution contained 140 mM KF, 10 mM HEPES, 5 mM CaCl_2_, 10 mM MgCl_2_, 55 mM EGTA, pH 7.22. To measure KCa1.1 currents, the intracellular solution contained 140 mM KCl, 5 mM HEPES, 9.8 mM CaCl_2_, 2mM MgCl_2_, 10 mM EGTA, pH 7.4 resulting in 4.5 µM free Ca^2+^ in the solution. To measure hErg1 currents the pipette solution contained 140 mM KCl, 10 mM HEPES, 2 mM MgCl_2_, 10 mM EGTA, pH 7.3., while the bath solution contained 140 mM choline chloride, 5 mM KCl, 10 mM HEPES, 20 mM glucose, 2 mM CaCl_2_, 2 mM MgCl_2_, 0.1 mM CdCl_2_, pH 7.35.

Astemizole, an open-channel blocker of hEag1 [36] was used to inhibit the hEag1 channels in SaOS-2. TEA (tetraethylammonium) and paxilline were used to inhibit the KCa1.1 K^+^ current. Blockers were applied directly on the studied cells using a microperfusion system (AutoMate Perfusion Pencil and Multi-Barrel Manifold Tip with 250 µm in diameter, AutoMate Scientific, Berkeley, CA, USA).

### 4.6. Calcium Deposition Assay

Calcium deposition was assayed with alizarin red S staining. The staining was carried out as previously described [18,31]. Alizarin red staining, although indicating Ca^2+^ deposits, is widely used to report on mineralization of various tissues including stem cells and cancer cells [18,64,65]. Cell monolayers were photographed after staining, and for the quantification, alizarin red S–calcium complexes were extracted from the stained cultures with 10% cetylpyridinium chloride in 10 mM sodium phosphate buffer (pH 7.7). The optical densityabsorbance of the extract was determined at 540 nm using spectrophotometer in each well. PBS and 0.1% (*v*/*v*) DMSO treatments were used as control. Four percent (*v*/*v*) DMSO induces hydroxyapatite formation [91], so data obtained with PBS and DMSO controls were statistically identical. Normalized Ca^2+^ deposition was calculated as A/A_PBS_ where A is the absorbance of a given sample and A_PBS_ is the average absorbance of differentiation-induced cells in the presence of PBS (vehicle control).

### 4.7. Cell Viability/Proliferation

Cell viability was determined with three different methods: methylthiazolyldiphenyl-tetrazolium bromide (MTT) assay, lactate dehydrogenase (LDH) cytotoxicity assay and sulphorodamine B (SRB) assay in “high density cultures” (see above, Materials and Methods 4.1). LDH assay is a colorimetric assay based on the measurement of lactate dehydrogenase (LDH) activity released from the cytosol of damaged cells into the supernatant. The assay was performed using Cytoscan LDH Cytotoxicity Assay (G-Biosciences) according to the manufacturer’s instructions. The SRB assay was carried out as described previously [92] with some modifications. Its principle is based on the ability of the negatively charged pink aminoxanthine dye, sulphorodamine B (SRB; Biotium) to bind to the basic amino acids in the cells [93]. The most common method for the determination of cell viability is the MTT reduction assay. The measurement is based on the conversion of MTT to formazan by cells in the culture. MTT assay (Sigma-Aldrich) was carried out according to the manufacturer’s instructions. PBS and DMSO treatments were used as vehicle controls, results were normalized to non-differentiated SaOS-2 cells in the presence of DMSO.

### 4.8. Data Analysis

For multiple comparisons, a one-way RM ANOVA with a post-hoc Holm–Sidak (HS) test were used. Statistical significance was concluded at *p* < 0.05 (* indicates statistical difference; *n* ≥ 3). The effect of the astemizole at a given concentration was reported as the remaining current fraction (RCF = *I*/*I*_0_, where I and I_0_ are current amplitudes in the presence and absence, respectively, of the inhibitor at a given concentration). Points on the dose–response curve represent the mean of 3–8 independent measurements where the error bars represent the SEM. Data points were fitted with a four-parameter Hill equation:(1)$$\frac{I}{I_{0}}=a+\frac{\left(1-a\right)}{1+{\left(\frac{x}{IC_{50}}\right)}^{n_{H}}}$$
where *IC*_50_ is the half-maximal inhibitory concentration of astemizole, *n_H_* is the Hill coefficient, and “*a*” is the residual, non-blocked current fraction.

K^+^ current traces were fitted with a single exponential function rising to the maximum according to the Hodgkin–Huxley model, I(t) = I_a_ × [1 − exp(−t/τ_act_)]^4^ + C, where I_a_ is the amplitude of the activating curve component, τ_act_ is the activation time constant of the current, and C is the amplitude of the non-activating current component. The τ_act_ for a particular cell was defined as the average of τ_act_ values obtained for at least three depolarizing pulses repeated at every 15 s in a sequence.

## Figures and Tables

**Figure 1 ijms-23-10533-f001:**
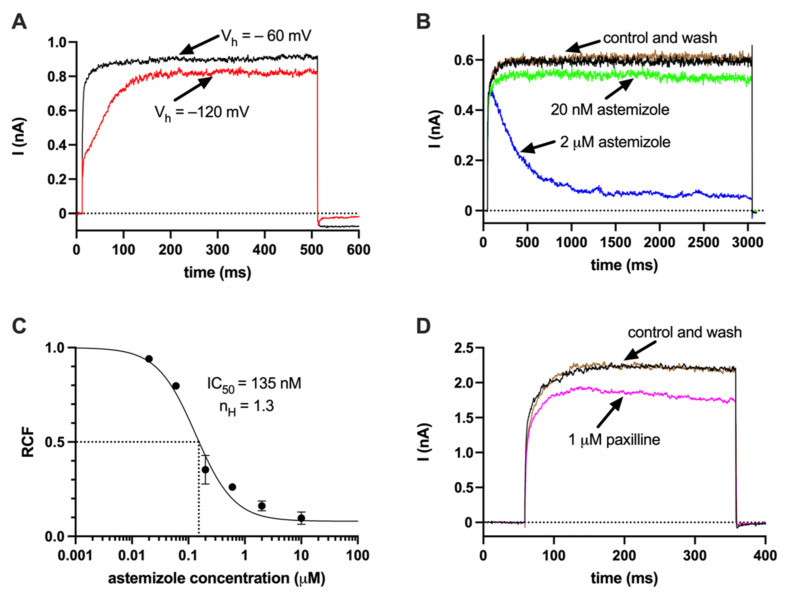
hEag1 and KCa1.1 currents in SaOS-2 cells (**A**) Activation kinetics of the current from different holding potentials. Whole-cell currents were recorded using two different holding potentials (Vh): −60 mV (black) and −120 mV (red). The depolarizing test pulses to +50 mV were 500 ms-long. (**B**) Inhibition of hEag1 currents by the open channel blocker astemizole. Astemizole (20 nM, green; 2 µM, blue) was administered in whole-cell configuration during 3000-ms-long test pulses to +50 mV from a holding potential of −60 mV, the inhibition was reversible (control: black; wash-out: brown) (**C**) Astemizole concentration–response curve. Six different doses of astemizole were tested as indicated. The remaining current fractions (RCF) were determined as *I*/*I*_0_, where *I* is the from the current at equilibrium block measured at the end of the 3000-ms-long depolarizing pulses, and I_0_ is the current in the absence of astemizole (see panel (**B**) for the demonstration of equilibrium block). (**D**) Paxilline-sensitive KCa1.1 current. Whole-cell current measurements were carried out using a 4.5 µM free Ca^2+^-containing intracellular solution and +100 mV test pulses (300 ms duration) to maximally activate the KCa1.1 K^+^ current. Traces were recorded in the absence (control, black), and in the presence of 1 µm paxilline (magenta) and following the wash-out (wash-out, green). The paxilline-sensitive current fraction was 17% in this cell.

**Figure 2 ijms-23-10533-f002:**
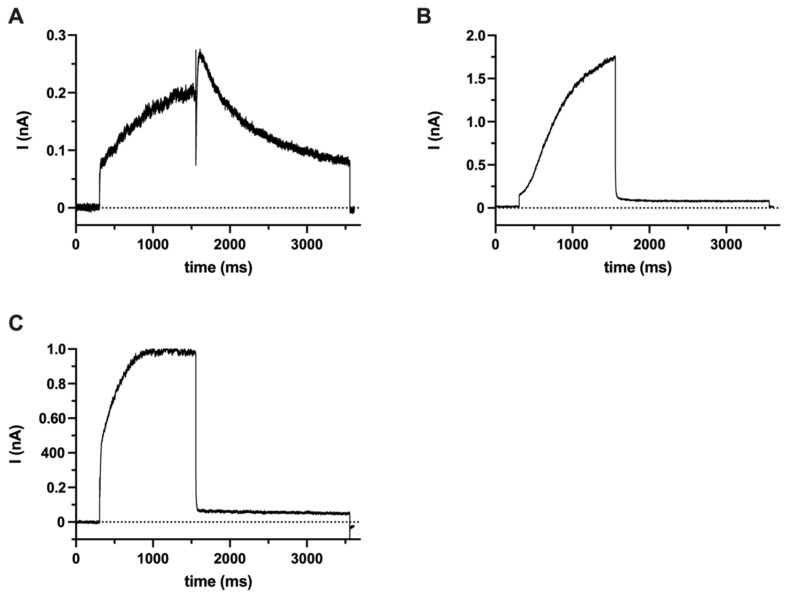
Lack of hErg1 currents in SaOS-2 cells (**A**) hErg1 whole-cell current was recorded using hErg1 solutions (see Methods for details) in HEK 293 cells stably expressing hErg1 (Kv11.1) gene. The measurements were carried out using a +20 mV test pulse (1250 ms-long) and followed by a 2000-ms-long test pulse to −40 mV. The holding potential was −80 mV. These measurements were repeated in hEag1(Kv10.1)-transfected HEK cells (**B**) and SaOS-2 cells (**C**) using the same experimental conditions and voltage protocol.

**Figure 3 ijms-23-10533-f003:**
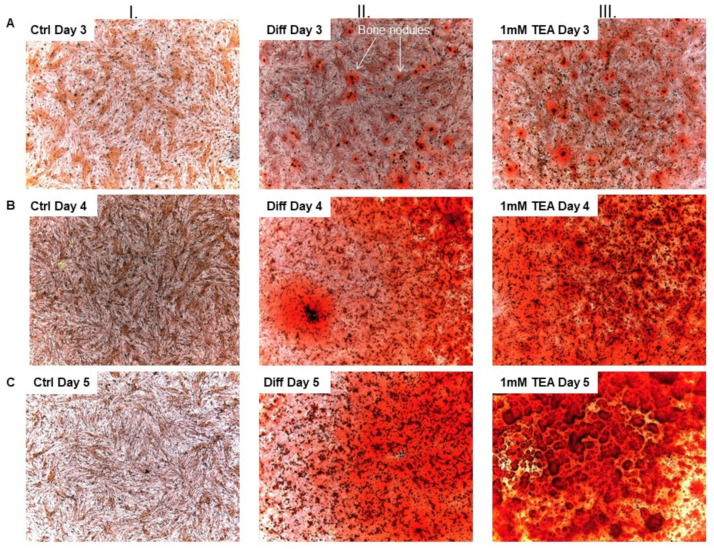
Ca^2+^ deposition induced by the classical pathway of osteogenesis in the presence of TEA in SaOS-2 cells. Ca^2+^ deposits were stained using alizarin red S (see Materials and Methods). Alizarin red S-stained cultures were photographed with a microscope equipped with a digital camera. The photos are presented according to the time of the examination by the rows and the type of treatment by the columns. Column I.: non-differentiated cell cultures (Ctrl, no differentiation cocktail) on Day 3 (row (**A**)), Day 4 (row (**B**)) and on Day 5 (row (**C**)). Column II.: differentiated cells (Diff, classical induction pathway cocktail) after the osteogenic induction in the same order as above. Column III.: differentiated cells in the presence of TEA (1 mM TEA, classical induction pathway cocktail+1 mM TEA) in the same order as above. The hydroxyapatite containing bone nodules are recognizable with the red color.

**Figure 4 ijms-23-10533-f004:**
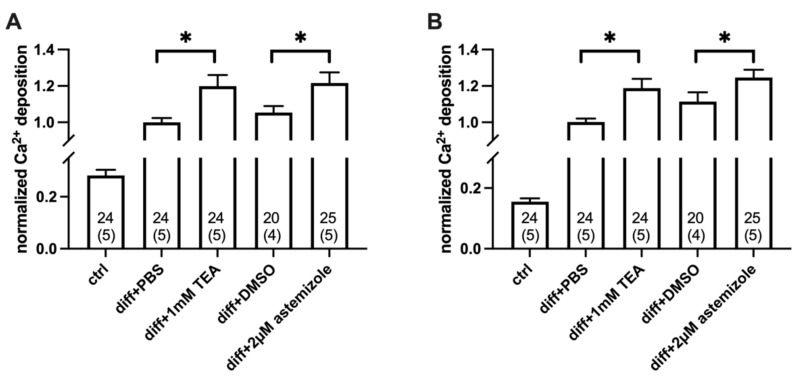
Classical pathway-induced Ca^2+^ deposition in SaOS-2 cultures in the presence and absence of TEA and the hEag1 inhibitor astemizole. TEA (1 mM) and astemizole (2 µM) were added to the cell cultures at the beginning of osteogenic induction. The Ca^2+^ deposits were detected using alizarin red staining and quantification was achieved by dissolving alizarin red–calcium complexes in CPC and measuring absorbance (see Materials and Methods). Normalized Ca^2+^ deposition was calculated as A/A_PBS_ where A is the absorbance of a given sample and A_PBS_ is the average absorbance of differentiation-induced cells in the presence of PBS (vehicle control). Labels indicate SaOS-2 cell cultures treated with ctrl: no differentiation cocktail; diff+PBS: induced by the classical pathway of osteogenesis and added PBS as vehicle control for TEA; diff+1 mM TEA: induced by the classical pathway and treated with 1 mM TEA; diff+DMSO: induced by the classical pathway and added DMSO as vehicle control for astemizole; diff+2 µM astemizole: induced by classical pathway of osteogenesis and treated with 2 µM astemizole. Data were obtained on Day 3 (**A**) and Day 4 (**B**). Data are presented as mean ±SEM (numbers in the bars indicate the number of data points and in parentheses the number of independent experiments) and analyzed using the one-way RM ANOVA statistical test, * *p* < 0.05.

**Figure 5 ijms-23-10533-f005:**
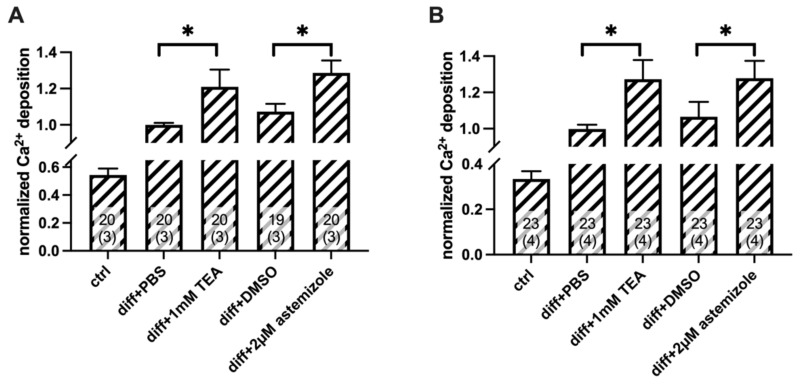
Pathological pathway (Pi)-induced Ca^2+^ deposition in SaOS-2 cultures in the presence and absence of TEA and the hEag1 inhibitor astemizole. Mineralization of SaOS-2 cells was induced via the pathological pathway using inorganic phosphate (Pi). Alizarin red–calcium complexes were determined using alizarin red assay on Day 3 (**A**) and Day 4 (**B**) (See Figure 4 and Methods for details). Normalized Ca^2+^ deposition was calculated as A/A_PBS_ where A is the absorbance of a given sample and A_PBS_ is the average absorbance of differentiation-induced cells in the presence of PBS (vehicle control). TEA and astemizole were added to differentiation-induced cultures in 1 mM and 2 µM concentrations, respectively, at the beginning of osteogenic induction. Labels indicate SaOS-2 cell cultures treated with ctrl: no differentiation induction by Pi; diff+PBS: differentiation induced by Pi and added PBS as vehicle control for TEA; diff+1 mM TEA: differentiation induced by Pi and treated with 1 mM TEA; diff+DMSO: differentiation induced by Pi and added DMSO as vehicle control for astemizole; diff+2 µM astemizole: differentiation induced by Pi and treated with 2 µM astemizole. Data are presented as mean ±SEM (numbers in the bars indicate the number of data points and in parentheses the number of independent experiments) and analyzed using the one-way RM ANOVA statistical test, * *p* < 0.05.

**Figure 6 ijms-23-10533-f006:**
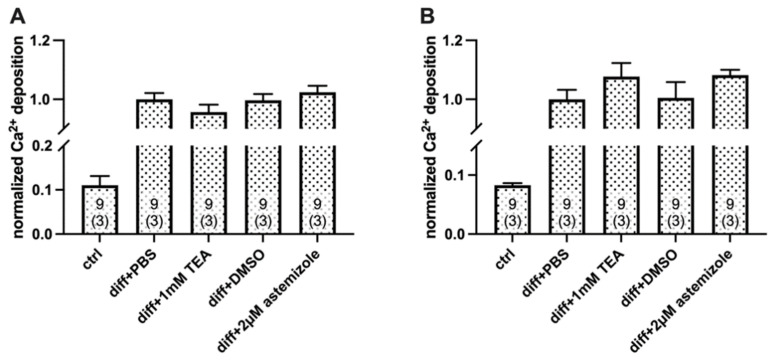
Delayed application of ion channel blockers to mineralization-induced SaOS-2 cells. Osteogenic differentiation of SaOS-2 cells was induced by the classical pathway and the ion channel blockers in the indicated concentrations were added 24 h after the induction. Alizarin red–calcium complexes were determined using alizarin red assay on Day 4 (**A**) and Day 5 (**B**) (See Figure 4 and Methods for details). Normalized Ca^2+^ deposition was calculated as A/A_PBS_ where A is the absorbance of a given sample and A_PBS_ is the average absorbance of differentiation-induced cells in the presence of PBS (vehicle control). Labels indicate SaOS-2 cell cultures treated with ctrl: no differentiation cocktail; diff+PBS: induced by the classical pathway of osteogenesis and added PBS as vehicle control for TEA; diff+1 mM TEA: induced by the classical pathway and treated with 1 mM TEA; diff+DMSO: induced by the classical pathway and added DMSO as vehicle control for astemizole; diff+2 µM astemizole: induced by classical pathway of osteogenesis and treated with 2 µM astemizole. Data are presented as mean ±SEM (numbers in the bars indicate the number of data points and in parentheses the number of independent experiments) and analyzed using One Way RM ANOVA statistical test.

**Figure 7 ijms-23-10533-f007:**
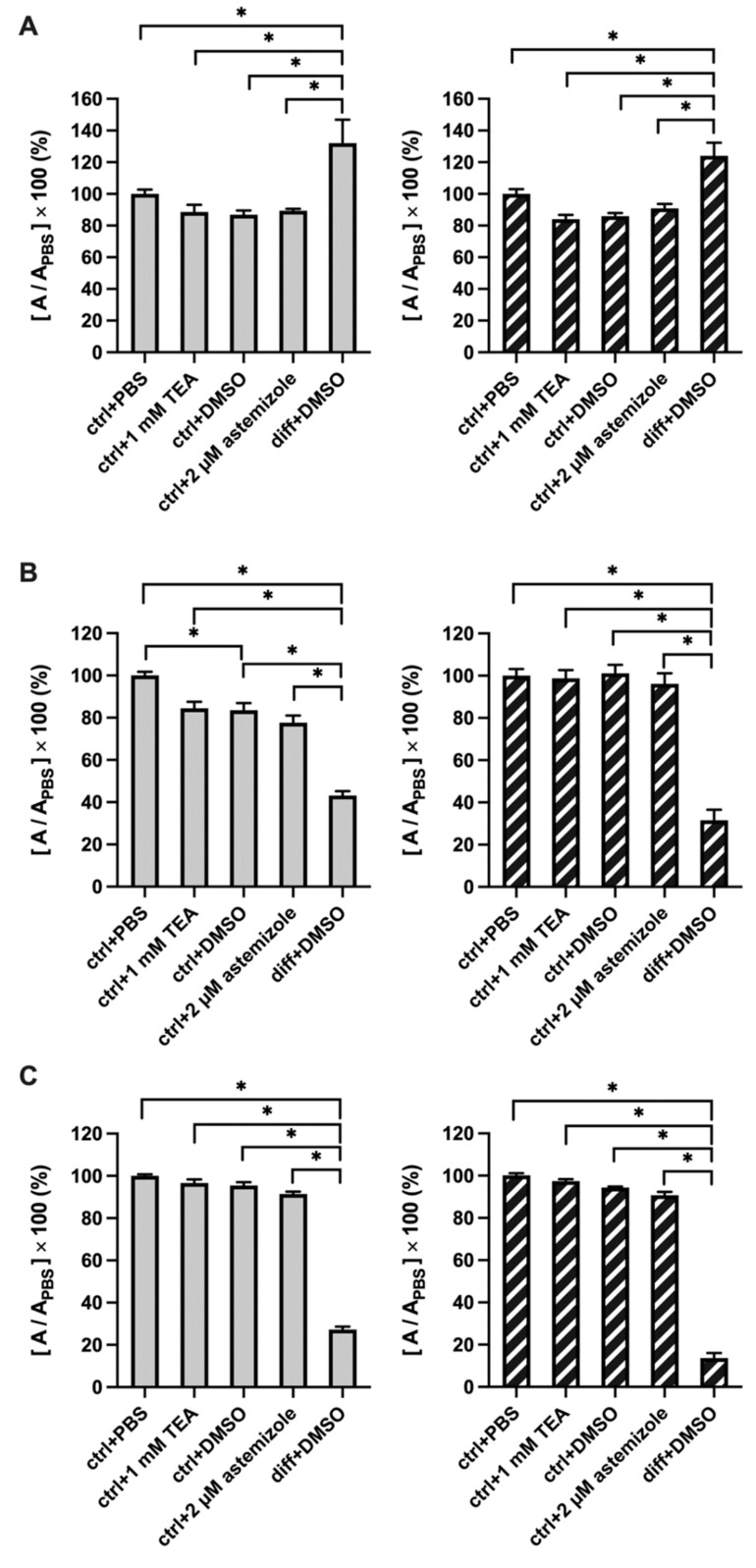
K^+^ channel blockers do not increase cytotoxicity. Normalized absorbance was calculated as [A/A_PBS_] × 100 (%), where A and A_PBS_ are the absorbances of a given sample and that of SaOS-2 cells in the presence of PBS control, respectively. Astemizole and TEA were added to the cell culture medium in the indicated concentrations. Absorbance determinations were carried out on samples harvested on Day 3 (gray columns) and Day 4 (hatched columns). Bar heights and error bars indicate mean ± SEM (*n* > 9). Data were analyzed using the one-way ANOVA statistical test, * *p* < 0.05. (**A**) LDH release assay. Labels indicate SaOS-2 cell cultures treated with ctrl+PBS: added PBS as vehicle control for TEA; ctrl+DMSO: added DMSO as vehicle control for astemizole; ctrl+1 mM TEA: treated with 1 mM TEA; ctrl+2 µM astemizole: treated with 2 µM astemizole; diff+DMSO: mineralization induced by the classical pathway and added DMSO as vehicle control for astemizole. (**B**) MTT assay. The same set of experiments as in panel A was repeated using MTT assay, treatments and variables are the same as above. Similarly, PBS treatment was used as vehicle control, results were normalized to SaOS-2 cells treated with PBS control. (**C**) SRB assay. The same set of experiments as in panels A and B were repeated using SRB assay, treatments and variables are the same as above. PBS treatment was used as vehicle control, results were normalized to SaOS-2 cells treated with PBS control.

## Data Availability

The data presented in this study are available on request from the corresponding author.

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
