# Peer review of "The hEag1 K^+^ Channel Inhibitor Astemizole Stimulates Ca^2+^ Deposition in SaOS-2 and MG-63 Osteosarcoma Cultures"

_ijms, 2022, doi:10.3390/ijms231810533_

Round 1

Reviewer 1 Report

In the manuscript by Mészáros et al, entitled “The hEag1 K+ channel inhibitor astemizole stimulates Ca2+ deposition in SaOS-2 and MG-63 osteosarcoma cultures”, the authors are interested by the role played by Eag1, a voltage-gated K+ channel, in the human osteosarcoma cell lines SaOS-2 and MG-63. They demonstrate that a K+ current corresponding to the expression of Eag1, but not to the closely related Erg1, can be recorded in SaOS-2 cells by using suitable electrophysiological protocols. They show that this Eag current can be inhibited by astemizole with an IC50 close to the value reported in the literature for this channel. They also show that the cells express a K+ current sensitive to paxilline, an inhibitor of KCa1.1, a voltage- and Ca-dependent K+ channel that is also sensitive to TEA. The authors show that blockade of Eag1 or KCa1.1 by astemizole or TEA, respectively, increases the Ca2+ accumulation in SaOS-2 and GM-63 cultures treated with a mineralization inducing cocktail. On the other hand, these channel blockers do not induce cytotoxicity.

I think that this study is quite interesting and brings some important clues to elucidate the functions of K+ channel in tumor cells. The experiments appear to have been carefully conducted. I think that the manuscript is of interest for a broad audience in International Journal of Molecular Sciences.

1. However, Eag-1 is found in almost all human tumors. Its expression in osteosarcoma cell lines SaOS-2 and GM-63 and its involvement in tumorigenicity has been previously studied. Some of these studies are indeed cited in the manuscript (réf 11, 21, 23, 24). It is therefore abusive to hypothesize “that hEag1 potassium channels may be present in SaOS-2 osteosarcoma cells” (Page 2 line 59). Moreover, targeting Eag channels with therapeutic agents has also been largely evaluated and I am not sure that astemizole would be the best candidate to play this role. In fact, astemizole is not used even as anti-histaminic in many countries, and is far to be “a specific open-channel blocker” (line 574)! Erg1 is 200X more sensitive to astemizole than Eag1 (Zhou et al (2007) PMID 10376921)! I agree that Eag is an important potential target for cancer therapy but I doubt about the simple repurposing of astemizole!

2. I think that the manuscript would be easier to read if it was indicated in the Introduction section (some of these elements appear only in the Discussion section):

- that the expression of Eag, KCa1.1 and Kv7.3 have been detected (RT-PCR, WB) in SaOS-2 and MG-63 cells (ref 11, 21, 23, 24).

- that inhibition of Eag expression by siRNA or shRNA decreases proliferation and migration in SaOS-2 and GM-63 (ref 11, 21)

- that KCa1.1 current and Kv7.3 current have been identified thanks to their pharmacological sensitivity to paxilline and TEA, and to linopirdine and XE991, respectively (ref 12, 13). But, Eag current have never been unambiguously identified.

- that the blockade of K+ current through KCa1.1 and Kv7.3 with these drugs increase Ca2+ deposition (as already stated line 54-58). But the effects produce by the blockade of Eag on Ca2+ deposition have not been investigated.

                3. The Supplementary Fig. 1 is unnecessary. It must at least been stated that this confirms earlier results.

                4. I do not understand the utility of the supplementary Fig. 2!

                5. The authors suggest that the residual current at high astemizole concentration should result from KCa1.1 expression. But, there is a residual current in Eag expressing HEK cells (Garcia-Ferreira et al (2004) PMID 15365094) or in Xenopus oocytes (Gomez-Varela et al (2004) PMID 16949586). Did the authors try to block the residual current by paxilline (in the presence of astemizole) ? Since the effects produced by 1mM TEA and 2microM astemizole on Ca2+ deposition are similar, is it possible to conclude that the K+ currents through Eag and KCa1.1 are similar during differentiation experiments? Electrophysiological experiments seem to reveal that the K+ current depends largely from Eag1 activity?

                6. The authors used TEA at 1mM to block KCa1.1, a concentration that induce 90% current inhibition (line 438), but a dose dependent block of KCa1.1 is not provided! Did the authors also used paxilline instead of TEA with a similar efficacy?

                7. Did the authors try to use simultaneously astemizole and TEA in the differentiation experiments?

                8. The authors conclude that the “K+ conductance is responsible for the observed effect” (line 441), i.e. the Ca2+ deposition, irrespective of the K+ channels. But I don’t understand why this conclusion is supported by the lack of effect if the blockers were applied 24h following induction of mineralization (line 442)?

                9. Previous studies have shown that the proliferation and migration of both SaOS-2 and GM-63 cells were affected by Eag1 inhibition. It seems to me that these effects are independent from differentiation and without induction of mineralization! I would like to get the authors‘ opinion on this?

Minor comments/typos

1. Paxilline (alkaloid) but not paxillin (Paired box protein Pax-1)

2. line 52 “ it sems”

3. line 55 “hHv1 proportion”

4. line 59 “we hypothesed”

5. line 68 “observed if of astemizole”

6. line 84 “this activation kinetics”

7. line 106 “based on these we conclude”

8. line 170 “activity of during differentiation”

9. line 180 “The microscopic photographs”

10. line 290 “TEA and astemizole does not induce”

11. line 357 “is characterized bay”

12. line 379 “The transcript…and the expression …was also reported”

Author Response

First of all, we would like to thank Reviewer 1 for the time and effort devoted to reviewing our manuscript, and we thank the critical comments and suggestions for the restructuring of the manuscript. Please find our itemized answers below:

  1. However, Eag-1 is found in almost all human tumors. Its expression in osteosarcoma cell lines SaOS-2 and GM-63 and its involvement in tumorigenicity has been previously studied. Some of these studies are indeed cited in the manuscript (réf 11, 21, 23, 24). It is therefore abusive to hypothesize “that hEag1 potassium channels may be present in SaOS-2 osteosarcoma cells” (Page 2 line 59). Moreover, targeting Eag channels with therapeutic agents has also been largely evaluated and I am not sure that astemizole would be the best candidate to play this role. In fact, astemizole is not used even as anti-histaminic in many countries, and is far to be “a specific open-channel blocker” (line 574)! Erg1 is 200X more sensitive to astemizole than Eag1 (Zhou et al (2007) PMID 10376921)! I agree that Eag is an important potential target for cancer therapy but I doubt about the simple repurposing of astemizole!

Answer:

We completely agree with Reviewer 1 that the expression of Eag1 has been shown in both SaOS-2 and MG-63 cells using molecular biology (PCR, Western Blot, Immune histochemistry). As Reviewer 1 pointed out, we also cite these references in the manuscript. We intended to claim in our hypothesis that hEag1 potassium channels may be functionally expressed in osteosarcoma cells. All the previous studies that we are aware of lacked the electrophysiological confirmation Eag-1 currents in SaOS-2. Accordingly, we have modified the corresponding statement in the manuscript to express what we intended to do, i.e., show the functional expression of the channels. In our view, showing the Eag-1 currents is the strongest evidence for the functional expression of the channels in SaOS-2.

As for the selectivity of astemizole and a potential repurposing, we must agree with the reviewer that these were strong statements and accordingly we have toned these down in the revised version. We have rewritten several paragraphs to express our view better, i.e., a potential mechanism of osteosarcoma treatment is to induce terminal differentiation by the inhibition of K+ channels. One candidate is targeting Eag-1 given that a more selective inhibitor is available.

  1. I think that the manuscript would be easier to read if it was indicated in the Introduction section (some of these elements appear only in the Discussion section):

- that the expression of Eag, KCa1.1 and Kv7.3 have been detected (RT-PCR, WB) in SaOS-2 and MG-63 cells (ref 11, 21, 23, 24).

- that inhibition of Eag expression by siRNA or shRNA decreases proliferation and migration in SaOS-2 and GM-63 (ref 11, 21)

- that KCa1.1 current and Kv7.3 current have been identified thanks to their pharmacological sensitivity to paxilline and TEA, and to linopirdine and XE991, respectively (ref 12, 13). But, Eag current have never been unambiguously identified.

- that the blockade of K+ current through KCa1.1 and Kv7.3 with these drugs increase Ca2+ deposition (as already stated line 54-58). But the effects produce by the blockade of Eag on Ca2+ deposition have not been investigated.

Answer: We are sincerely thankful to Reviewer 1 for the constructive critiques. We have incorporated these ideas into the introduction. We must agree, now the introduction gives a better background and motivation for the experiments.

  1. The Supplementary Fig. 1 is unnecessary. It must at least been stated that this confirms earlier results.

Answer: Thank you for the suggestion, we have indicated that Suppl Fig 1 is a confirmation of earlier experiments.

  1. I do not understand the utility of the supplementary Fig. 2!

Answer: Our aim here was to demonstrate and confirm that both MG-63 and SaOS-2 cells express Eag1 transcripts, albeit at different extent. The lower expression of Eag1 in MG-63 parallels with the slower kinetics of mineral matrix production in MG-63 as compared to SaOS-2. However, we feel that this might simply be a co-incidence and we have not shown the causal relationship between Eag1 expression and mineralization kinetics. As such, we removed Suppl Fig 2. Thank you for the suggestion.

  1. The authors suggest that the residual current at high astemizole concentration should result from KCa1.1 expression. But, there is a residual current in Eag expressing HEK cells (Garcia-Ferreira et al (2004) PMID 15365094) or in Xenopus oocytes (Gomez-Varela et al (2004) PMID 16949586). Did the authors try to block the residual current by paxilline (in the presence of astemizole) ? Since the effects produced by 1mM TEA and 2microM astemizole on Ca2+ deposition are similar, is it possible to conclude that the K+ currents through Eag and KCa1.1 are similar during differentiation experiments? Electrophysiological experiments seem to reveal that the K+ current depends largely from Eag1 activity?

Answer:

Thank you very much for the comments, and the information about astemizole block. We agree completely with Reviewer 1 that there seems to be a residual current Eag1 current, much more in oocytes (PMID 16949586) than in HEK cells (PMID 15365094), nevertheless, the dose-dependence of the Eag1 inhibition by astemizole could be well described by a Boltzmann function in HEK without residual current (the fitted curves in Fig 1 in the Garcia-Ferreira paper were run between 0 and 1 without obvious offset). In our case, however, the contribution of an astemizole-resistant current fraction had to be included in the fit, this fraction is remarkably larger than the astemizole-resistant current in HEK cells (PMID 15365094). This residual current can be KCa1.1 or Kv7.3, as pointed out by Reviewer 1 above and also mentioned in the discussion of the manuscript. We did not apply paxilline after astemizole block in our experiments as we were focusing on the confirmation of the presence of Eag1 in the cells, whereas the functional expression of KCa1.1 (“MaxiK”) has already been shown in these cells (J Membr Biol. 1996 Mar;150(2):175-84. doi: 10.1007/s002329900042). Panel D in Fig. 1 illustrates that one component of the remaining current can be KCa1.1, it is functional expression is simply demonstrated by that panel without quantitative determination of the TEA affinity.

To what extent the individual currents contribute to the transmembrane currents of a living cell is extremely difficult to predict: the composition of the intra- and extracellular solutions and the voltage protocols are optimized for recording of a given ion current. These optimal conditions may never be reached under physiological conditions, i.e., very positive membrane potentials at which these currents were recorded are unlikely to be achieved in a classically non-excitable cell. However, based on the similarity in the sensitivity of Ca2+ deposition to 2 microM astemizole and 1 mM TEA we agree with the conclusion of Reviewer 1 that although there might be more Eag1 channels in the membrane these osteosarcoma cells (~larger current) the K+ currents conducted by Eag1 and KCa1.1 may regulate Ca2+ deposition in a similar manner.

  1. The authors used TEA at 1mM to block KCa1.1, a concentration that induce 90% current inhibition (line 438), but a dose dependent block of KCa1.1 is not provided! Did the authors also used paxilline instead of TEA with a similar efficacy?

The reviewer is right, we have not measured the TEA+ affinity of the KCa1.1 current. Rather, we have used data in the literature (reference 68 in the original manuscript, now also added original papers to the Kd estimate) and we assumed that the affinity of channel for extracellular TEA remains the same in SaOS-2 cells. It is known that the affinity of peptide toxins varies with the beta subunit composition of the KCa1.1 channel complex, e.g. beta4 in complex with the KCa1.1 alpha subunit makes the channel resistant to IbTX (PMID: 10792058), but we are unaware of such beta subunit dependence of the TEA affinity. Therefore, we assumed that the TEA affinity does not change. We have not used paxillin in the in vitro Ca2+ deposition experiments since we found that TEA is very easy and safe to use in cell cultures.    

  1. Did the authors try to use simultaneously astemizole and TEA in the differentiation experiments?

Thank you for the question, we have not combined the two treatments. The combination of the two drugs may have additive effect on the increase in the Ca2+ deposition. We did not investigate this scenario in this manuscript, we focused on the consequences of the  inhibition of Eag1 on Ca2+ deposition, and used TEA data as an important reference point.

  1. The authors conclude that the “K+ conductance is responsible for the observed effect” (line 441), i.e. the Ca2+ deposition, irrespective of the K+ channels. But I don’t understand why this conclusion is supported by the lack of effect if the blockers were applied 24h following induction of mineralization (line 442)?

Thank you very much for pointing out this issue. You are absolutely right; we have corrected the paragraph. The optimal time window for the blockers may only indicate that there is an early signaling event which is sensitive to the K+ conductance of the membrane.

  1. Previous studies have shown that the proliferation and migration of both SaOS-2 and GM-63 cells were affected by Eag1 inhibition. It seems to me that these effects are independent from differentiation and without induction of mineralization! I would like to get the authors‘ opinion on this?

We agree with the note of Reviewer 1, is seems that proliferation, migration and differentiation of the studied osteosarcoma cells may be controlled by Eag1 in a distinctly different manner. We would like to point out here that we used high density cultures in this study which are ideal for differentiation induction rather than for studying cell proliferation and migration. Thus, we cannot exclude the possibility that other factors, such as inhibition of proliferation and migration might influence the overall outcome of the Ca2+ deposition response upon K+ channel inhibition. It may be that inhibition of migration may “concentrate” the cells to form a “high density environment” that may promote mineralization. Thank you very much for the motivating comment, we will consider this scenario in our subsequent studies.

Minor comments/typos

  1. Paxilline (alkaloid) but not paxillin (Paired box protein Pax-1). Thank you, apologies for this terrible mistake
  2. line 52 “ it sems”
  3. line 55 “hHv1 proportion”
  4. line 59 “we hypothesed”
  5. line 68 “observed if of astemizole”
  6. line 84 “this activation kinetics”
  7. line 106 “based on these we conclude”
  8. line 170 “activity of during differentiation”
  9. line 180 “The microscopic photographs”
  10. line 290 “TEA and astemizole does not induce”
  11. line 357 “is characterized bay”
  12. line 379 “The transcript…and the expression …was also reported”

Answer:

We have corrected all typos, thank you very much for listing them and making the corrections easier and more efficient. We also apologize for them, especially the paxilline vs paxillin issue.

Finally, we thank Reviewer 1 the helpful comments. We hope that Reviewer 1 will find the revised version of the manuscript improved and suitable for publication.

Reviewer 2 Report

The major contribution of this work is to characterize the functional Kv10.1 in SaOS-2 and MG1 osteosarcoma cultures. The study is potentially interesting, however, more control experiments are required to make the conclusion.

Major:

1. Mainly, whether it is Kv10.1 but not other members in the family mediates the current. Whether Kv10.1 needs to form heterotetramer with other members, for example Kv10.2 or Kv2.1 to mediate this current. The expression levels of other members of Kv10 and Kv11 need to be tested in SaOS-2 and MG1 osteosarcoma cultures by RT-PCR. Testing Kv11.1 alone is not enough. 

2. Since the inhibitor is not specific, it is important to know whether the current can be attenuated by down-regulation of Kv10.1 via Crispr KO or RNAi. If there is other subunit that forms heterotetramer with Kv10.1, whether down-regulation of this subunit can decrease the current as well.

Minor:

The spelling needs to be checked more carefully. For example, line 52 "it sems that..." should be corrected. 

Author Response

We would like to thank Reviewer 2 for the time and effort allocated to reviewing our manuscript. We have addressed the critical comments below

Major:

  1. Mainly, whether it is Kv10.1 but not other members in the family mediates the current. Whether Kv10.1 needs to form heterotetramer with other members, for example Kv10.2 or Kv2.1 to mediate this current. The expression levels of other members of Kv10 and Kv11 need to be tested in SaOS-2 and MG1 osteosarcoma cultures by RT-PCR. Testing Kv11.1 alone is not enough. 
  2. Since the inhibitor is not specific, it is important to know whether the current can be attenuated by down-regulation of Kv10.1 via Crispr KO or RNAi. If there is other subunit that forms heterotetramer with Kv10.1, whether down-regulation of this subunit can decrease the current as well.

Answer: We feel that both points 1 and 2 are related to whether it is indeed Kv10.1 currents that were measured and if astemizole had targeted this channel to produce the observed biological response.

Indeed, Kv10.1 can heteromultimerize with Kv10.2. The homotetrameric Kv10.1 and Kv10.2 currents are quite different, Kv10.2 mediated currents have significantly slower activation kinetics. Moreover, in Kv10.1/Kv10.2 heterotetramers the slow time course of eag2 (Kv10.2) activation dominates (Schönherr et al, FEBS Lett. 2002 Mar 13;514(2-3):204-8. doi: 10.1016/s0014-5793(02)02365-7. Bauer and Schwarz, J Physiol 596.5 (2018) pp 769–783, PMID: 29333676.) Moreover, our recording solution contained 1 mM Mg2+ which further accentuates the difference between Kv10.1 and Kv10.2. We used a single exponential term to describe the activation kinetics of the current in SaOS-2 cells, the time constant for the activation kinetics was ~67 ms at -100 mV holding using a +50 mV test potential. This agrees well with the slow component of the activation kinetics (~60-70 ms) measured for Kv10.1 in CHO cells by Schönherr et al using -100 mV holding potential and +50 mV test potential. Using the same pulse protocol activation time constant for the homotetrameric Kv10.2 channel is ~300 ms. As the activation kinetics of the heterotetramers is dominated by the Kv10.2 (Schönherr et al), we do not think that the measured currents were heteromeric Kv10.1/Kv10.2 ones.

As for and Kv2.1/Kv10.1 heterotetramers, we do not think that they might exist. The papers which proposed their existence used a misleading nomenclature (PMID: 15046870 DOI: 10.1016/j.molbrainres.2004.01.004; PMID: 12060745 doi: 10.1073/pnas.122617999); what they call as Kv10.1 is KCNG3 encoding the Kv6.3 channel, their Kv11.1 is KCNV2 encoding Kv8.2 and their Kv6.3 is KCNG4 encoding Kv6.4. To our knowledge, there is no evidence showing heteromultimer formation between the real Kv10.1 (encoded by the KCNH1gene) and Kv2.1 (Luis Pardo, personal communication). Moreover, Kv10.1 does not make heterotetramers with Kv11.1 either (Luis Pardo, personal communication).

Our molecular biology (Suppl fig 2, revised version numbering) and electrophysiology (Fig. 2) data strongly argues for the lack of Kv11.1 in SaOS-2 cells. As Kv11.1, Kv11.2 and Kv11.3 currents share common kinetic features (Shi et al, The Journal of Neuroscience, December 15, 1997, 17(24):9423–9432), we prefer to use the same kinetic argument against the functional expression of Kv11.2 and Kv11.3 that we used against the expression of Kv11.1 in the manuscript.

Based on these we would prefer to sustain our conclusion that the currents shown in Fig. 1a are Kv10.1 ones.  We hope that our arguments above will convince Reviewer 2 about our claims in the manuscript, and Reviewer 2 will find the manuscript acceptable in the IJMS.

Minor:

The spelling needs to be checked more carefully. For example, line 52 "it sems that..." should be corrected.

Answer: Thank you very much, we have corrected multiple spelling errors in the manuscript, we deeply regret those.

Round 2

Reviewer 2 Report

The author made a good augment. The manuscript is acceptable in IJMS.